# Diverse enteric bacterial, viral, and parasitic pathogen genes are shed in animal feces in Indiana

Anna A. Heintzman[1], Ishi Keenum[2], Drew Capone[1*]

**1** Department of Environmental and Occupational Health, Indiana University, Bloomington, Indiana, United States of America, **2** Department of Civil, Environmental, and Geospatial Engineering, Michigan Technological University, Houghton, Michigan, United States of America

* dscapone@iu.edu

## Abstract

Southern Indiana has intensive livestock production, yet species-resolved fecal pathogen and pathogen associated gene profiles are limited. At 10 sites in southern Indiana (April–June 2024), we collected 128 fecal specimens from 10 hosts: pigs (n = 12), horses (12), cats (12), chickens (12), dogs (22), white-tailed deer (12), sheep (12), goats (12), cows (12), and humans (10). We extracted and assayed total nucleic acids using a custom 43-target TaqMan Array Card (RT-qPCR). Flotation microscopy was performed on pig and dog stools for helminth ova. In-silico specificity checks were conducted for selected targets due to potential for cross reactivity between pathogen species. Most samples (60%, 75/126) were positive for ≥1 target, including enteropathogenic *Escherichia coli* (*eae*) 16% and shiga toxin genes (*stx1* 10%, *stx2* 6.3%). Higher prevalence of genes associated with specific pathogens and gut microbes in specific animals was common, including *E. coli* O157:H7 in pigs (42%) and sheep (8.3%); *Campylobacter coli* in chickens (36%) and *Klebsiella pneumoniae* in humans (60%) and dogs (9.1%). We found the protozoa *Giardia* in 15% of samples (notably dogs 32%, cows 33%) and *Cryptosporidium* in 14% (cats 55%, cows 25%, chickens 27%). Most (55%) chicken samples were positive for *Plasmodium*, which aligned with evidence of locally circulating avian haemosporidians. The *Ascaris lumbricoides* assay was positive only in pigs (17%), and we identified *Ascaris* type eggs in 92% of pig samples via microscopy, suggesting our *Ascaris lumbricoides* assay cross reacted with *Ascaris suum* supporting detection of the swine lineage (*A. suum*). We detected the class 1 integron-integrase gene (*intI1*) in 43% of stools, concentrated in chickens, pigs, and horses. These findings suggest animal feces poses a public health hazard in Southern Indiana and indicate the need for targeted One Health studies to better understand the public health risks of specific exposures and animal feces management practices (e.g., farm storage capacity, land application timing, soil incorporation/injection, tile-drain proximity).

**Data availability statement:** All relevant data are within the paper and its Supporting Information files.

**Funding:** This work was supported by the National Science Foundation (Award # 2412444 to DC). This research was also funded by the Indiana University Advanced Summer Research Scholarship (2024 award to AH). The funders had no role in study design, data collection and analysis, decision to publish, or preparation of the manuscript.

**Competing interests:** The authors have declared that no competing interests exist.

## Introduction

Enteric pathogens and their genes, including pathogenic bacteria, viruses, protozoa, and helminths are shed in the feces of infected humans and animals [1]. Exposure to and infection by pathogenic species contributes to the global burden of disease [2,3]. Many enteric organisms of public health concern originate from animal reservoirs, where they persist and often circulate asymptomatically, particularly among livestock and wildlife populations [4–6]. Enteric pathogens may be transmitted to humans through direct or indirect contact, including via soil, water, food, flies, and fomites [7]. In agricultural and peri-domestic settings, where humans live in close proximity to animals and their waste, shared spaces for waste disposal, food production, and water access can create high-risk interfaces for zoonotic transmission. Mitigating exposure and interrupting transmission pathways likely requires a holistic, multisectoral approach to disease surveillance and control, consistent with One Health principles focused on zoonoses, antimicrobial resistance, and food safety [8–11].

Previous research documented the enteric pathogens present in animal fecal matter [4,12]. Molecular studies further indicate that these bacteria, viruses, protozoa, and helminths commonly co-occur in animal feces and often overlap with those found in human populations, suggesting shared environmental reservoirs and transmission pathways [13]. Recent evidence also highlights the fecal dissemination of antimicrobial-resistant (AMR) organisms, such as extended-spectrum β-lactamase (ESBL)-producing *Escherichia coli*, from animals to humans in low- and middle-income country (LMIC) settings [14–16]. Despite increased attention, the full extent of microbial and resistance gene diversity across animal hosts—including livestock, companion animals, and wildlife—remains insufficiently characterized. Species-specific analysis of these targets could guide One Health interventions targeting zoonotic diseases, AMR, and food safety risks.

Southern Indiana represents an area with intensive livestock farming, yet there is a paucity of data regarding enteric pathogen and pathogen associated gene profiles among animal species. Indiana is home to approximately 8.2 million broiler chickens, 36 million layer chickens, 8.5 million turkeys, 4.4 million hogs and pigs, and 770,000 cattle and calves, far exceeding the state's human population of roughly 6.8 million (2022 data) [17]. Combined, these Indiana livestock generate 18.8 billion kg of feces annually, compared to approximately 320 million kg produced by the state's human population (S1 Table) [18]. Adequate containment and treatment of these fecal wastes is necessary to limit environmental fecal contamination and zoonotic exposure risk. The objective of this study was to characterize enteric bacterial, viral, protozoan, and helminth profiles via detection of pathogen genes in fecal specimens collected from 10 host species, including humans, in agricultural and rural environments in southern Indiana during April-June 2024. The findings will contribute to a broader understanding of the potential health risks posed by exposure to animal feces in this and similar settings.

## Materials and methods

### Study sites and species of interest

We selected ten species for stool sample collection including pigs (*Sus scrofa domesticus*), horses (*Equus caballus*), cats (*Felis catus*), chickens (*Gallus gallus domesticus*), dogs (*Canis lupus familiaris*), white-tailed deer (*Odocoileus virginianus*), sheep (*Ovis aries*), goats (*Capra aegagrus hircus*), cows (*Bos taurus*), and humans (*Homo sapiens*). Stool samples were purposively collected from 10 sites across Southern Indiana. We obtained verbal permission from business and landowners before animal fecal collection.

128 stool samples were collected between April and June, 2024 from 10 sites across Newberry, Solsberry, Freedom, and Bloomington, Indiana. Sample counts by species were as follows: pigs (n = 12), horses (n = 12), cats (n = 12), chickens (n = 12), dogs (n = 22), deer (n = 12), sheep (n = 12), goats (n = 12), cows (n = 12), and humans (n = 10). No formal sample size calculations were performed, because our aim was to broadly assess profiles of enteric pathogens and their genes in animal feces.

### Sample collection protocol

We collected animal fecal specimens noninvasively from fresh droppings using sterile 10 µL inoculating loops directly into sterile 2 mL cryogenic tubes (Thermo Fisher Scientific, Waltham, MA). Samples were stored at 4°C for <24 hours, and then frozen at −80°C until processing. Species identity was confirmed based on direct observation, input from the landowner, and contextual information (e.g., farm location, enclosures). We collected cat feces from litter boxes, dog feces from bagged waste in dog-park trash cans, and horse feces from stall bedding. All other animal feces samples were collected off the ground. Human stool samples were self-collected by adult volunteers under IRB protocol #18634, using the Fisherbrand™ Commode Specimen Collection System (Thermo Fisher Scientific, Waltham, MA) and instructions. Written informed consent was obtained from all human participants prior to sample collection.

### Nucleic acid extraction

We conducted extractions in a Class II Type A2 biological safety cabinet that was decontaminated before and after each use with 10% bleach and 70% ethanol followed by ultraviolet (UV) light exposure. Consumables used during extraction were either pre-sterilized, single-use items opened only within the biosafety cabinet, or reusable materials that had been washed and autoclaved prior to use.

We performed extractions of total nucleic acids from stool using the QIAamp 96 Virus QIAcube HT Kit (Qiagen, Hilden, Germany), following a protocol adapted from the manufacturer's *QIAcube HT Handbook* (Qiagen, 2015). The procedure included a manual pretreatment step followed by automated extraction using the QIAcube HT instrument [19,20].

Approximately 100 mg of fecal material was transferred into PowerBead Pro Tubes containing 1 mL of Buffer ASL (Qiagen, Hilden, Germany) and 10 µL of Inforce 3 vaccine (Zoetis, Parsippany, NJ) containing bovine respiratory syncytial virus (BRSV) and bovine herpesvirus (BHV) as an exogenous positive control. Tubes were weighed before and after sample addition to record the precise fecal mass. Samples were homogenized using a TissueLyser II (Qiagen, Hilden, Germany) at 25 Hz for two 5-minute cycles, with a 180° rotation of the tube block between cycles. Lysates were centrifuged at 14,000 rpm for 2 minutes to pellet debris and incubated at room temperature for 10–15 minutes before extraction. Supernatants (200 µL) were transferred into S-block wells and subjected to automated extraction using the QIAcube HT with the manufacturer's standard viral nucleic acid purification protocol. Extracts were eluted in 100 µL of AVE buffer and stored at −80°C.

We included one negative extraction control (no sample added, Inforce 3 vaccine only) in each extraction batch. The spiked Inforce 3 vaccine served as an internal positive control for downstream qPCR. DNA concentrations were measured using the Qubit dsDNA High Sensitivity Assay Kit (Thermo Fisher Scientific, Waltham, MA) on the Qubit 4 Fluorometer. Input masses, dsDNA yields from nucleic acid extraction, and equivalent sample mass used in qPCRanalysis are summarized in S2 Table.

Following extraction, each purified template was further processed using the Zymo Research OneStep™ PCR Inhibitor Removal Kit (Zymo Research, Cat. #D6030) following the manufacturer's instructions. Our DNA extraction protocol had been optimized for human feces and the inhibitor removal kit may have helped remove the various PCR inhibitors potentially present in animal feces.

## qPCR pre-screening and gene target detection via custom TaqMan Array Card (TAC)

We used qPCR to assay all samples for BRSV to assess potential inhibition, given that our methods were previously optimized for human and not animal feces. We tested the undiluted template and template diluted 1:10. Reactions were prepared using the AgPath-ID™ One-Step RT-PCR Kit (Thermo Fisher Scientific, Waltham, MA). We included a reverse transcription step at 45°C for 20 minutes, followed by an initial denaturation at 95°C for 10 minutes. Amplification consisted of 45 cycles of 95°C for 15 seconds and 60°C for 1 minute, with a ramp rate of 1°C/second between all steps, which was run on a Quantstudio 7 Instrument (Thermo Fisher Scientific, Waltham, MA).

We developed a custom TaqMan Array Card (TAC) according to Liu *et al.* 2013 [21] and 2016 [22] for the simultaneous detection of multiple enteric pathogen and gut microbe genes (S3 Table, S4 Table). The gut pathogen-associated targets assessed included helminths (*Ancylostoma duodenale*, *Ascaris lumbricoides*, *Necator americanus*, *Schistosoma mansoni*, *Strongyloides stercoralis*, and *Trichuris trichiura*); protozoa (*Cryptosporidium* spp., *Entamoeba histolytica*, *Giardia* spp., and *Plasmodium* spp.); bacteria (*Campylobacter coli*, *Clostridium difficile*, *Escherichia coli* O157:H7, enteroaggregative *E. coli*, enteropathogenic *E. coli*, enterotoxigenic *E. coli*, *Helicobacter pylori*, *Klebsiella pneumoniae*, *Plesiomonas shigelloides*, *Salmonella enterica* serovar Typhi, *Salmonella* spp., *Shigella* spp./enteroinvasive *E. coli*, shiga-toxin producing *E. coli*, *Vibrio cholerae*, and *Yersinia enterocolitica*); and viruses (Adenovirus, Astrovirus, bovine herpesvirus, bovine respiratory syncytial virus, Influenza A, Norovirus GI, Norovirus GII, respiratory syncytial virus, Rotavirus, Sapovirus, and SARS-CoV2). The TAC platform also included a molecular target for *Candida auris*.

The card incorporated several quality control assays: a 16S rRNA assay [23]; a human mitochondrial DNA assay to confirm host DNA presence; and the class 1 integron-integrase gene (*intI1*) as an indicator of antimicrobial resistance potential.

For each TAC run, a reaction mastermix was prepared using the AgPath-ID™ One-Step RT-PCR Kit (Thermo Fisher Scientific, Waltham, MA). For each sample, 60 µL of this mastermix was combined with 40 µL of 1:10 diluted nucleic acid extract (or control). All nucleic acid extracts analyzed on TAC were diluted 1:10 in nuclease-free water to minimize PCR inhibition, because we observed similar Cq values for BRSV in 1:10 diluted and undiluted samples for some animal species in our previously described qPCR analysis for BRSV. On each day of TAC analysis, one no-template control (NTC), consisting of 40 µL of nuclease-free water and 60 µL of mastermix, was included to monitor for potential contamination.

Thermal cycling was performed on a QuantStudio™ 7 Flex Real-Time PCR System (Applied Biosystems) using cycling parameters from Liu et al. 2013 [21] and 2016 [22]. The protocol included a reverse transcription step at 45°C for 20 minutes, followed by an initial denaturation at 95°C for 10 minutes. Amplification consisted of 45 cycles of 95°C for 15 seconds and 60°C for 1 minute, with a ramp rate of 1°C/second between all steps.

The TAC performance was verified using an eight-point, 10-fold serial dilution ($10^2$–$10^9$ gene copies per reaction) of two plasmids containing all gene targets and constructed log-linear standard curves. The plasmid containing DNA targets was diluted and used directly. The plasmid containing RNA targets was linearized (BshT I enzyme), transcribed using the MEGAscript™ T7 Transcription Kit, and the transcribed product was cleaned using the MEGAclear™ Transcription Clean-Up Kit (Thermo Fisher Scientific, Waltham, MA). Of the 46 candidate assays, 44 met accepted quality criteria (linearity ≥ 0.97, efficiency: 90–120%), with efficiencies spanning 98–120% (median = 106%), linearities of 0.992 to 0.999, slopes of –3.63 to –2.98, and y-intercepts of 33.4–40.8. Two assays, the leukocyte and adenovirus 40/41 assays, did not perform well on the standard curve, and they were therefore excluded from downstream analyses. Additionally, although the RSV assay met performance criteria on the standard curve, we used bovine respiratory syncytial virus (BRSV) as an internal positive control, and subsequent testing indicated cross-reactivity between the RSV primers/probe and BRSV sequences (93% identity across

the probe and >80% identity across primers; GenBank AF295544.1). This resulted in spurious detections of RSV in multiple samples. Because of this cross-reactivity, we did not include RSV results in the final dataset. Summary performance statistics for the remaining 43 assays (slope, efficiency, y-intercept, and R²) are presented in S4 Table.

All TAC data were analyzed in QuantStudio™ Real-Time PCR Design & Analysis Software 2.8.0. Quantification cycle (Cq) values were determined using manual thresholding and interpreted relative to standard curves and daily NTCs (S1 Fig). Targets were categorized as negative if amplification did not occur or occurred beyond a Cq of 40, in order to reduce the risk of false-positive results due to background signal or nonspecific amplification (S5 Table).

### Helminth egg detection via microscopy

Microscopy was performed on fecal samples from pigs and dogs to investigate the potential for cross-reactivity between morphologically similar helminth species. Approximately 1 gram of feces was combined with 18 mL of sodium nitrate flotation solution (specific gravity = 1.22) in a commercial fecal flotation device equipped with a built-in manual mixing rod and screw-top lid. Samples were homogenized, and approximately 1 mL of suspension was transferred to each reading chamber of a Mini-FLOTAC device, as described in Cringoli *et al.* 2017 [24]. After a 10-minute flotation period, the rotating coverslip was aligned for microscopic viewing. All samples were examined under light microscopy at 100× magnification for the presence of helminth eggs, which were confirmed by morphological characteristics and size.

### In silico analysis of TAC assay primers and probes

The analytical specificities of primers and probes for the *Ancylostoma duodenale*, *Ascaris lumbricoides*, *Trichuris trichuria*, and *Plasmodium* spp. assays were evaluated in silico using Primer–BLAST (NCBI, accessed 21 August 2025) against the nt database, restricted to *Ancylostoma*, *Ascaris*, *Trichuris*, and *Plasmodium*, respectively. Pairwise alignment of *A. duodenale* (MK271367.1) with *A. ceylanicum* (PP527745.1) and *A. caninum* (MT130933.1); *A. lumbricoides* (PP758217.1) with *A. suum* (PP758228.1); *T. trichiura* (LC800551.1) with *T. vulpis* (HF586909.1) and *T. ovis* (HF586911.1); and *P. falciparum* (MN852864.1) with *P. relictum* (PV628049.1) sequences was performed with BLASTN ('Align two sequences').

## Results

### Controls

There was no amplification of any pathogen-associated targets in the eleven negative extraction controls, except for consistent positive signals in the BHV (100%, 11/11) and BRSV (100%, 11/11) assays. Among the twelve TAC no-template controls, no targets showed amplification below the positivity threshold (Cq < 40). We detected mastermix contamination in five TAC cards, but all affected samples were re-run with a new lot of reagents and the contamination was eliminated. We observed amplification of BRSV in all samples (128/128), while BHV was detected in 98.4% of samples (126/128).

We included several additional quality control assays on the TAC to assess sample composition and quality. The 16S rRNA assay amplified in 100% of fecal samples (128/128). We detected human mitochondrial DNA—a common fecal source tracking assay [25]—in 100% of human samples (10/10, sensitivity = 100%) and 6.8% of other samples (8/118, specificity = 93%).

Two specimens that failed to amplify the BHV extraction/amplification control (one cat and one chicken sample) were excluded from subsequent analyses. The analyses presented herein therefore include the 126 specimens that were positive for both BHV and 16S rRNA. Each of the original 128 specimens yielded amplification of at least one internal control, and we provide the complete dataset in S6 Table.

### Molecular detection of enteric pathogen-associated targets

Of the 126 specimens analyzed, 75 (60%) showed amplification for at least one pathogen-associated target. Bacterial targets were the most frequently detected target group. Detection prevalence for each TAC molecular target (by host species

and overall) is summarized in Table 1. Enteropathogenic *E. coli* (*eae*) was the most prevalent *E. coli* pathotype, detected in 16% (20/126) of samples and across 8 species, with the highest prevalence in sheep (50%, 6/12). We detected *E. coli* O157:H7 only in pigs (42%, 5/12) and sheep (8.3%, 1/12), while we found enterotoxigenic *E. coli* (LT and STp) exclusively in pigs (17%, 2/12) and deer (8.3%, 1/12), respectively. No samples tested positive for *Shigella* spp./enteroinvasive *E. coli*, though we detected STEC (stx1) and STEC (stx2) in 10% (13/126) and 6.3% (8/126) of samples, respectively, primarily from sheep.

Other notable bacterial detections included *Campylobacter coli* (n = 4, all in chickens), and *Klebsiella pneumoniae*, found in 60% (6/10) of human samples and 9.1% (2/22) of dog samples. We found *intl1*, an antimicrobial resistance marker, in 43% (54/126) of samples, with especially high prevalence in chickens (100%, 11/11), pigs (83%, 10/12), and horses (83%, 10/12).

*Giardia* spp. was the most prevalent protozoan, which we detected in 16% (19/126) of samples, especially among dogs (32%, 7/22) and cows (33%, 4/12). We detected *Cryptosporidium* spp. in 13% (17/126) of samples, with the highest prevalence in cats (55%, 6/11), cows (25%, 3/12), and chickens (27%, 3/11). We detected *Plasmodium* spp. in 7.9% (10/126) of samples, most commonly in chickens (55%, 6/11) and cows (17%, 2/12).

Among helminths, we only detected *Ascaris lumbricoides* in pigs (17%, 2/12). We did not detect *Ancylostoma duodenale*, *Necator americanus*, *Schistosoma mansoni*, *Strongyloides stercoralis*, or *Trichuris Trichuria* in any samples.

We detected norovirus GII and astrovirus infrequently (0.8%, 1/126), and did not detect any other viral targets, including norovirus GI, rotavirus, adenovirus, and SARS-CoV-2, in any samples.

We analyzed 28 samples run in duplicate. Across these samples, we identified 17 distinct pathogen genes for which at least one replicate tested positive. In the 28 samples, there were 166 instances in which at least one replicate was positive for a given target. We observed perfect positive concordance (i.e., both duplicates tested positive) in 79% (n = 131/166) of replicates. Conversely, in the 28 samples, there were 1,213 instances in which at least one replicate was negative for a given target. We observed perfect negative concordance (i.e., both duplicates tested negative) in 97% (n = 1,178/1,213) of replicates. Overall agreement across all duplicate comparisons (i.e., total matches/total comparisons) was 97% (n = 1,309/1,344).

## Microscopy

We performed microscopy on fecal samples from pigs and dogs to investigate possible cross-reactivity. We did not observe whipworm eggs (*Trichuris* spp.) in any dog samples (0/22), consistent with the absence of *Trichuris trichiura* detections by qPCR. In contrast, we found eggs morphologically consistent with *Ascaris* spp. in nearly all pig samples (92%, 11/12), supporting the TAC detections of *Ascaris lumbricoides*. Though this suggests likely cross-reactivity with *Ascaris suum*, which commonly infects swine. S7 Table summarizes the quantitative microscopy data, listing (i) the mass of stool used to prepare each NaNO$_3$ flotation suspension, (ii) the resulting stool concentration of that suspension (g mL$^{-1}$), (iii) the number of eggs counted in a single 0.5 mL Mini-FLOTAC chamber, and (iv) the calculated eggs-per-gram (EPG) of stool.

## In silico analysis

In silico analysis demonstrated that the ITS2 region of *Ancylostoma duodenale* (MK271367.1) shared 98% identity with *A. ceylanicum* (PP527745.1), with a single nucleotide mismatch in the probe binding site (S2 Fig). Alignment with *A. caninum* (MT130933.1) showed 98% identity and four total mismatches across primers and the probe (S2 Fig). We also found that the ITS1 region of *Ascaris suum* (PP758228.1) is 99% identical to *A. lumbricoides* (PP758217.1), with only a single nucleotide difference in the probe binding site (S3 Fig). The 18S rRNA region of *T. trichuria* (LC800551.1) is 93% identical to *T. vulpis* (HF586909.1) and 95% identical to *T. ovis* (HF586911.1), each with a single nucleotide mismatch in forward primer binding site (S4 Fig). The 18S rRNA region of *Plasmodium falciparum* (MN852864.1) is 91% identical to *P. relictum* (PV628049.1) with no mismatches in either the primer or probe binding sites (S5 Fig).

**Table 1. Prevalence of enteric microbial, parasitic, and antimicrobial-resistance gene targets detected by custom TAC RT-qPCR in fecal samples from 10 host species collected at 10 sites in southern Indiana, April–June 2024.**

| Type | Target | \multicolumn{11}{c}{Prevalence (%; number positive/total samples)} | | | | | | | | | | |
| | | Cat | Chicken | Cow | Deer | Dog | Goat | Horse | Pig | Sheep | Human | Total |
|---|---|---|---|---|---|---|---|---|---|---|---|---|
| Virus | Astrovirus | 0.0% (0/11) | 0.0% (0/11) | 0.0% (0/12) | 0.0% (0/12) | 0.0% (0/22) | 0.0% (0/12) | 8.3% (1/12) | 0.0% (0/12) | 0.0% (0/12) | 0.0% (0/10) | 0.8% (1/126) |
| | BHV | 100% (11/11) | 100% (11/11) | 100% (12/12) | 100% (12/12) | 100% (22/22) | 100% (12/12) | 100% (12/12) | 100% (12/12) | 100% (12/12) | 100% (10/10) | 100% (126/126) |
| | Norovirus GII | 0.0% (0/11) | 0.0% (0/11) | 0.0% (0/12) | 0.0% (0/12) | 0.0% (0/22) | 0.0% (0/12) | 0.0% (0/12) | 0.0% (0/12) | 0.0% (0/12) | 10% (1/10) | 0.8% (1/126) |
| Bacteria | *Campylobacter jejuni/ coli* | 0.0% (0/11) | 36% (4/11) | 0.0% (0/12) | 0.0% (0/12) | 0.0% (0/22) | 0.0% (0/12) | 0.0% (0/12) | 0.0% (0/12) | 0.0% (0/12) | 0.0% (0/10) | 3.2% (4/126) |
| | *Clostridioides difficile* | 0.0% (0/11) | 0.0% (0/11) | 0.0% (0/12) | 0.0% (0/12) | 4.5% (1/22) | 0.0% (0/12) | 0.0% (0/12) | 0.0% (0/12) | 0.0% (0/12) | 0.0% (0/10) | 0.8% (1/126) |
| | Enteropathogenic *E. coli* (bfpA) | 0.0% (0/11) | 0.0% (0/11) | 0.0% (0/12) | 0.0% (0/12) | 0.0% (0/22) | 0.0% (0/12) | 0.0% (0/12) | 8.3% (1/12) | 17% (2/12) | 0.0% (0/10) | 2.4% (3/126) |
| | Enteropathogenic *E. coli* (eae) | 9.1% (1/11) | 0.0% (0/11) | 0.0% (0/12) | 17% (2/12) | 18% (4/22) | 25% (3/12) | 8.3% (1/12) | 167% (2/12) | 50% (6/12) | 10.0% (1/10) | 16% (20/126) |
| | Enterotoxigenic *E. coli* (LT) | 0.0% (0/11) | 0.0% (0/11) | 0.0% (0/12) | 0.0% (0/12) | 0.0% (0/22) | 0.0% (0/12) | 0.0% (0/12) | 17% (2/12) | 0.0% (0/12) | 0.0% (0/10) | 1.6% (2/126) |
| | Enterotoxigenic *E. coli* (STp) | 0.0% (0/11) | 0.0% (0/11) | 0.0% (0/12) | 8.3% (1/12) | 0.0% (0/22) | 0.0% (0/12) | 0.0% (0/12) | 0.0% (0/12) | 0.0% (0/12) | 0.0% (0/10) | 0.8% (1/126) |
| | *Escherichia coli* O157:H7 | 0.0% (0/11) | 0.0% (0/11) | 0.0% (0/12) | 0.0% (0/12) | 0.0% (0/22) | 0.0% (0/12) | 0.0% (0/12) | 42% (5/12) | 8.3% (1/12) | 0.0% (0/10) | 4.8% (6/126) |
| | *Klebsiella pneumoniae* | 0.0% (0/11) | 0.0% (0/11) | 0.0% (0/12) | 0.0% (0/12) | 9.1% (2/22) | 0.0% (0/12) | 0.0% (0/12) | 0.0% (0/12) | 0.0% (0/12) | 60% (6/10) | 6.3% (8/126) |
| | *Plesiomonas shigelloides* | 27% (3/11) | 0.0% (0/11) | 0.0% (0/12) | 0.0% (0/12) | 0.0% (0/22) | 0.0% (0/12) | 0.0% (0/12) | 0.0% (0/12) | 0.0% (0/12) | 0.0% (0/10) | 2.4% (3/126) |
| | Shiga-toxin producing *E. coli* (stx1) | 0.0% (0/11) | 18% (2/11) | 0.0% (0/12) | 8.3% (1/12) | 0.0% (0/22) | 8.3% (1/12) | 0.0% (0/12) | 17% (2/12) | 58% (7/12) | 0.0% (0/10) | 10% (13/126) |
| | Shiga-toxin producing *E. coli* (stx2) | 0.0% (0/11) | 0.0% (0/11) | 0.0% (0/12) | 0.0% (0/12) | 0.0% (0/22) | 8.3% (1/12) | 0.0% (0/12) | 0.0% (0/12) | 58% (7/12) | 0.0% (0/10) | 6.3% (8/126) |
| | *Yersinia enterocolitica* | 0.0% (0/11) | 0.0% (0/11) | 0.0% (0/12) | 0.0% (0/12) | 0.0% (0/22) | 0.0% (0/12) | 0.0% (0/12) | 8.3% (1/12) | 0.0% (0/12) | 0.0% (0/10) | 0.8% (1/126) |
| Protozoa | *Cryptosporidium* spp. | 55% (6/11) | 27% (3/11) | 25% (3/12) | 0.0% (0/12) | 18% (4/22) | 0.0% (0/12) | 0.0% (0/12) | 8.3% (1/12) | 0.0% (0/12) | 0.0% (0/10) | 14% (17/126) |
| | *Giardia* spp. | 9.1% (1/11) | 9.1% (1/11) | 33% (4/12) | 0.0% (0/12) | 32% (7/22) | 8.3% (1/12) | 17% (2/12) | 0.0% (0/12) | 25% (3/12) | 0.0% (0/10) | 15% (19/126) |
| | *Plasmodium* spp. | 0.0% (0/11) | 55% (6/11) | 17% (2/12) | 0.0% (0/12) | 0.0% (0/22) | 8.3% (1/12) | 0.0% (0/12) | 8.3% (1/12) | 0.0% (0/12) | 0.0% (0/10) | 7.9% (10/126) |
| Helminth | *Ascaris lumbricoides* | 0.0% (0/11) | 0.0% (0/11) | 0.0% (0/12) | 0.0% (0/12) | 0.0% (0/22) | 0.0% (0/12) | 0.0% (0/12) | 17% (2/12) | 0.0% (0/12) | 0.0% (0/10) | 1.6% (2/126) |
| Control | 16S rRNA | 100% (11/11) | 100% (11/11) | 100% (12/12) | 100% (12/12) | 100% (22/22) | 100% (12/12) | 100% (12/12) | 100% (12/12) | 100% (12/12) | 100% (10/10) | 100.0% (126/126) |
| | Class 1 Resistance Integron (RI) | 0.0% (0/11) | 100% (11/11) | 17% (2/12) | 8.3% (1/12) | 14% (3/22) | 42% (5/12) | 83% (10/12) | 83% (10/12) | 75% (9/12) | 30% (3/10) | 43% (54/126) |
| | Human mtDNA | 9.1% (1/11) | 9.1% (1/11) | 8.3% (1/12) | 8.3% (1/12) | 0.0% (0/22) | 8.3% (1/12) | 17% (2/12) | 8.3% (1/12) | 0.0% (0/12) | 100% (10/10) | 14% (18/126) |

Prevalence is shown as % (positive/total) by host species and overall. Samples failing internal controls were excluded from prevalence calculations as described in the manuscript. Targets are reported as detected/not detected based on study positivity thresholds (Cq < 40 with manual thresholding). "Target" denotes detection of organism marker gene. Genes from the following pathogens were not detected in any samples: *Ancylostoma duodenale*, *Necator americanus*, *Schistosoma mansoni*, *Strongyloides stercoralis*, *Trichuris trichiura*, *Entamoeba histolytica*, enteroaggregative *E. coli*, enterotoxigenic *E. coli* (STh), *Helicobacter pylori*, *Salmonella enterica* serovar Typhi, *Salmonella* spp., *Shigella* spp./enteroinvasive *E. coli*, *Vibrio cholerae*, influenza A, norovirus GI, rotavirus, sapovirus, SARS-Cov2, *Candida auris*. TAC = TaqMan Array Card. Cq = quantification cycle. RT-qPCR = Reverse-Transcription Quantitative Polymerase Chain Reaction.

## Discussion

Our multi-species survey of feces in southern Indiana demonstrates that livestock, companion animals, wildlife, and humans harbor a diverse range of enteric pathogen genes. While bacterial targets were most common, we also detected protozoa pathogens (*Giardia* and *Plasmodium* spp.), helminths (*Ascaris* spp.), and viral pathogens (astrovirus and norovirus GII).

### Pathogen gene-specific findings

DNA from the genus *Plasmodium* was amplified in 8% of samples, including 50% of chicken samples. A recent study of wild songbirds in southern Indiana documented endemic haemosporidian infections and provided the first genetic evidence of avian-malaria lineages circulating in the state [26]. Outbreaks in captive penguins at Indiana zoos further indicate that transmission is occurring locally among avian hosts [27]. Because our qPCRassay targets a conserved mitochondrial locus, it cannot resolve parasite lineages, but the poultry signals most likely reflect avian *Plasmodium* and potential transmission from wild birds frequenting barns. This interpretation is supported by in silico alignments, which showed 100% identity of the primers and probe with *P. relictum*, a common avian malaria parasite. The 18S rRNA region of *P. falciparum*, a parasite that causes human malaria, also showed complete matches across all primer and probe binding sites (S5 Fig). Genomic sequencing and coordinated mosquito–bird surveillance would clarify lineage identity and transmission routes.

Only pig samples were positive for the *Ascaris lumbricoides* assay, and microscopy confirmed the presence of *Ascaris* type eggs in nearly all pig samples. Eggs of *A. suum* and *A. lumbricoides* are morphologically indistinguishable, and multiple genetic studies show the taxa are extremely closely related, with some studies treating them as a single species or a host-adapted complex. Importantly, zoonotic transmission of *Ascaris* spp. from pigs to humans has been documented in the United States, including 14 farm-associated cases in Maine and a recent autochthonous infection in a child living on a Mississippi pig farm [28,29]. Indiana residents in close contact with swine could face similar exposure risks. In addition, in silico analysis of the assay revealed that the *A. suum* ITS1 sequence differs from *A. lumbricoides* by only a single nucleotide within the probe binding site (S3 Fig). Assays designed for *A. lumbricoides* could amplify *A. suum* or fail to detect depending on the location of the mismatch in the primer and probe sequence [30–33]. Discordance between microscopy and TAC likely reflects the probe-template mismatch we found *in silico*. Closely matched sequences demonstrate the need for lineage-specific primers and probes (or sequencing of PCR amplicons) and an orthogonal method such as fecal flotation to corroborate molecular results when monitoring targets such as *Ascaris* across hosts.

Our in silico analyses also highlighted challenges for species-level resolution of *Ancylostoma* and *Trichuris*. The ITS2 region of *A. duodenale* shared 98% identity with *A. ceylanicum* and *A. caninum*, with mismatches occurring in only a few bases of the primer and probe binding sites (S2 Fig). Given this level of conservation, primers designed for *A. duodenale* may amplify other congeners, complicating interpretation of genus-level signals in mixed-host settings. Similarly, the 18S rRNA region of *T. trichiura* showed 93–95% identity to *T. vulpis* and *T. ovis*, with a single mismatch in the forward primer binding site (S4 Fig). This suggests that assays designed for human whipworm may also detect closely related congeners in dogs and ruminants, again underscoring the difficulty of attributing genus-level signals to a single host species. This potential for species cross reaction has generated debate about Blackburn *et al*. 2024 which reported qPCR based detection of *Trichuris trichiura* and *Ancylostoma duodenale* in soil samples from the United States [34,35].

Our detections of *Cryptosporidium* and *Giardia* could have public health implications. Although multiple assemblages and species circulate, only a subset are known to infect humans [36,37]. Because our genus-level assays cannot resolve lineages, species-level attribution is not possible. Nonetheless, *Cryptosporidium* detections in cattle most likely reflect *C. parvum*, a zoonotic lineage and leading cause of human cryptosporidiosis [38]. Likewise, *Giardia* assemblages A and B, which infect humans, are known to circulate in livestock and companion animals and can be transmitted to people through direct contact or contaminated water [39].

Bacterial targets were the most frequently detected pathogen group in our study. Many of the assays represent virulence-associated genes, including *eae* (enteropathogenic *E. coli*), *stx1* and *stx2* (Shiga toxin–producing *E. coli*), *LT* and *STp* (enterotoxigenic *E. coli*), and *cadF* (*Campylobacter jejuni/coli*), that are well established causes of diarrheal disease in humans and animals and are transmissible zoonotically [40–42]. These organisms may cause symptomatic or asymptomatic infections in animal hosts, but their shedding in feces creates potential for transmission to humans. Outbreaks of pathogenic *E. coli* linked to contaminated food products remain common in the United States, and livestock feces may contribute to the burden of foodborne diseases if wastes are not adequately managed [43]. The detection of *Campylobacter coli* in chickens is particularly concerning given that backyard poultry have repeatedly been implicated in human campylobacteriosis outbreaks [44]. In settings where people and animals interact closely, especially in peri-domestic or smallholder environments, basic hygiene measures such as handwashing are therefore critical in reducing zoonotic transmission.

The identification of *Klebsiella pneumoniae* in both human and dog samples also highlights the potential for bacterial pathogen exchange between humans and companion animals. Genomic studies have documented shared strains between dogs and their owners, indicating that interspecies transmission, while less common than human-to-human spread, does occur [45]. These findings emphasize that animal feces constitute a reservoir of bacterial pathogen-associated targets of recognized concern for human health in Southern Indiana, transmissible through both direct contact and contamination of the food chain.

The class 1 integron-integrase gene (*intI1*) was present in 43% of samples, with the highest prevalence in chicken, horse, pig, and sheep livestock. *intI1* commonly co-occurs with mobile resistance gene cassettes and is widely used as a proxy for anthropogenic antibiotic resistance gene (ARG) loading in water, soil, and fecal matrices [46–48]. The low prevalence of *intI1* we observed in human feces (30%) is similar to the reported prevalence among Swedish students [49]. Its distribution in our dataset underscores the importance of surveillance and waste management across human and animal sectors under a One Health framework [11].

Our study relied on qPCR to detect predefined genes associated with specific microorganisms [50]. PCR requires prior knowledge of the target sequence and cannot reveal other DNA that may be present. While metagenomic approaches could profile microbial communities more broadly, they generally have higher limits of detection than PCR and depend on reference databases for accurate taxonomic assignment [51].

It is also important to note that the TAC panel we used in this study was originally designed and validated for human stool. Applying these assays to mixed environmental or animal matrices can obscure taxonomic identity and complicate source attribution. Our in silico analyses of *Plasmodium*, *Ascaris*, *Ancylostoma*, and *Trichuris* highlight how these assays may cross-amplify closely related congeners, including zoonotic lineages. Without lineage or host-specific markers, microbial source-tracking assays, or sequencing of PCR amplicons, positive signals in environmental samples risk being misattributed to the wrong microbial species [35]. Complementary methods (*e.g.*, sequencing PCR products) may be helpful in future studies to minimize misclassification.

Beyond the assay-specific caveats noted above, our study has several additional limitations. First, the dataset is based on convenience sampling from one region during a single season, so temporal and geographic variation in gut pathogen gene prevalence could not be assessed. Second, due to the relatively high limit of detection, our analyses may have missed low-abundance targets. Third, we did not assess infectivity, so the organisms detected may or may not have been viable. Finally, while we analyzed a large overall number of samples, the number of samples per animal species was small, and thus our results may not be representative of all populations across Indiana.

## Policy and local context

Indiana's intensive livestock production creates opportunities for enteric pathogens to move from animal feces into soils, tile drainage, and surface waters, with downstream risks via drinking-, recreational-, and food-water pathways. Zoonotic

protozoa, helminths, and AMR markers across multiple hosts suggests a One Health approach to surveillance may be useful in this setting.

Regulatory oversight in Indiana is divided between the Indiana Department of Environmental Management (IDEM) and the Office of Indiana State Chemist (OISC). Large operations (≥ 300 cattle, 600 swine, 30,000 poultry, or 500 horses) operate under IDEM's Confined Feeding Operation (CFO) rule (327 IAC 19). This rule is designed primarily for nutrient control and water-quality protection and requires site-specific permits, land-application setbacks, nutrient-management plans, and enough storage capacity to avoid emergency application under adverse conditions, described as ≥180 days of no-discharge storage capacity (i.e., a capacity requirement, not a mandated retention time before land application) [52]. Sub-threshold operations and most manure uses outside CFOs are regulated by OISC's fertilizer-use rule (355 IAC 8), which closely mirrors CFO staging and field-application regulations (e.g., setbacks, covered/contained staging >72 h, and monitoring tile outlets during and immediately after application) [53]. OISC also administers licensing and certification for handlers of CFO manure (355 IAC 7). Compliance under these fertilizer rules is largely self-certified and typically enforced in response to complaints, and de minimis users (<10 yd³ or <4,000 gal per year) are exempt.

Neither framework requires routine microbiological monitoring of pathogens once manure is applied to fields. While prolonged storage can reduce viable loads for many organisms, persistence is expected for some taxa (e.g., *Ascaris* eggs), and because the ≥ 180-day standard is a storage-capacity requirement rather than a mandated residence time, manure may be applied well before pathogen die-off occurs. Consequently, pathogen release can still occur via lagoon overtopping during extreme precipitation, runoff from uncovered stockpiles, and transport through tile-drainage networks into receiving waters [54,55]. These wastes may pose occupational hazards for farm workers and their families and community risks through contact with contaminated surface waters during recreation. Future work is necessary to provide an empirical basis for evaluating whether additional, routine microbiological surveillance or control interventions are warranted within Indiana's existing regulatory framework that could reduce public health risks.

Our study demonstrates that animal feces in Southern Indiana contains enteric pathogens that pose a potential public health hazard. Additionally, applying qPCR assays validated for human stool to environmental and animal samples introduces important interpretive limitations, particularly at the genus level where cross-reactivity may occur. These findings highlight the public health hazards present in feces from pet and livestock populations, and also emphasize the caveats of applying qPCR assays beyond their original validation context, warranting cautious interpretation of positive signals. Because agricultural, peri-domestic, and wildlife interfaces clearly harbor overlapping pathogen and resistance reservoirs, coordinated One Health surveillance that incorporates host, vector, and environmental sampling, along with antimicrobial-use stewardship, will be critical for interrupting transmission events and mitigating downstream health impacts.

## Supporting information

**S1 Table. Estimated annual fecal mass production by Indiana livestock and humans, based on 2022 inventory/ population data and standard manure production coefficients (Indiana, USA).** Estimates reflect (i) annual fecal mass per individual animal or person, (ii) total state inventory/population, and (iii) calculated annual fecal mass produced (kg/ year).
(PDF)

**S2 Table. Nucleic-acid extraction and template input characteristics for fecal samples collected from 10 host species at 10 sites in southern Indiana, April–June 2024.** Table summarizes stool input mass (mg), double stranded DNA (dsDNA) concentration/yield (Qubit), and the calculated equivalent stool mass represented in each TaqMan Array Card (TAC) reaction following dilution and loading procedures.
(PDF)

**S3 Table. Primer and probe sequences for the custom TAC RT-qPCR assays used to screen fecal samples from 10 host species collected at 10 sites in southern Indiana, April–June 2024.** For each assay, the table lists target organism/marker, gene/region, and oligonucleotide sequences ($5' \rightarrow 3'$) for forward primer, reverse primer, and hydrolysis probe. TAC = TaqMan Array Card. RT-qPCR = Reverse-Transcription Quantitative Polymerase Chain Reaction.
(PDF)

**S4 Table. Standard-curve performance characteristics for assays included on the custom TAC RT-qPCR panel used to detect enteric microbial and parasitic nucleic-acid targets in fecal samples from southern Indiana, April–June 2024.** Table reports slope, y-intercept, $R^2$, and calculated amplification efficiency for each assay based on an eight-point 10-fold dilution series. Limit of detection (LOD) for assay validation was 100 gene copies/µL. TAC = TaqMan Array Card.
(PDF)

**S1 Fig. Representative amplification and multicomponent plots from custom TAC RT-qPCR runs used to detect enteric microbial and parasitic targets in fecal samples collected across 10 sites in southern Indiana, April–June 2024.** Plots illustrate typical positive and negative amplification patterns used for target calling in QuantStudio analysis. TAC = TaqMan Array Card. Cq = quantification cycle. RT-qPCR = Reverse-Transcription Quantitative Polymerase Chain Reaction.
(PDF)

**S5 Table. MIQE checklist for the custom TAC RT-qPCR workflow used to detect enteric microbial and parasitic targets in fecal samples collected across 10 sites in southern Indiana, April–June 2024.** Checklist items indicate where required methodological and reporting elements are addressed in the manuscript and supporting information. MIQE = Minimum Information for Publication of Quantitative Real-Time PCR Experiments. TAC = TaqMan Array Card. RT-qPCR = Reverse-Transcription Quantitative Polymerase Chain Reaction.
(PDF)

**S6 Table. Prevalence of *all* enteric microbial, parasitic, and antimicrobial-resistance gene targets detected by custom TAC RT-qPCR in *all* fecal samples from 10 host species collected at 10 sites in southern Indiana, April–June 2024.** Table includes results for all samples, including specimens later excluded from prevalence analyses due to internal-control performance, as described in the manuscript. Targets are reported as detected/not detected based on study positivity thresholds (Cq < 40 with manual thresholding). The following pathogenic targets were not detected in any samples: *Ancylostoma duodenale*, *Necator americanus*, *Schistosoma mansoni*, *Trichuris trichiura*, *Entamoeba histolytica*, enteroaggregative *E. coli*, enterotoxigenic *E. coli* (*STh*), *Helicobacter pylori*, *Salmonella enterica* serovar Typhi, *Salmonella* spp., *Shigella* spp./enteroinvasive *E. coli*, *Vibrio cholerae*, Influenza A, Norovirus GI, Rotavirus, Sapovirus, SARS-Cov2, *Candida auris*. TAC = TaqMan Array Card. Cq = quantification cycle. RT-qPCR = Reverse-Transcription Quantitative Polymerase Chain Reaction.
(PDF)

**S7 Table. Quantitative Mini-FLOTAC microscopy summary of pig and dog fecal samples collected at 10 sites in southern Indiana, April–June 2024.** Table shows stool mass used to prepare each $NaNO_3$ flotation suspension, resulting suspension concentration (g mL$^{-1}$), egg counts per 0.5 mL Mini-FLOTAC chamber, and calculated eggs-per-gram of stool (EPG).
(PDF)

**S2 Fig. In silico alignment of *Ancylostoma duodenale* (GenBank MK271367.1) with (left) *Ancylostoma ceylanicum* (PP527745.1) and (right) *Ancylostoma caninum* (MT130933.1) sequences in the assay region.** Forward primer

(positions 589–611) and reverse primer (638–659) are shaded in yellow and blue, respectively; the hydrolysis probe (619–636) is shaded in green. A single nucleotide mismatch at probe position 625 (boxed in red) distinguishes *A. duodenale* from *A. ceylanicum*, while three mismatches at probe positions 625, 633, and 636, as well as one mismatch as reverse primer position 644 distinguish *A. duodenale* from *A. caninum*.
(PDF)

**S3 Fig. In silico alignment of *Ascaris lumbricoides* (GenBank PP758217.1) and *Ascaris suum* (PP758228.1) sequences in the assay region.** The forward primer (113–137, yellow), reverse primer (225–246, blue), and probe (174–195, green) are all highlighted. A single nucleotide mismatch at probe position 178 (boxed in red) distinguishes *A. suum* from *A. lumbricoides*.
(PDF)

**S4 Fig. In silico alignment of *Trichuris trichiura* (GenBank LC800551.1) with (left) *Trichuris vulpis* (HF586909.1) and (right) *Trichuris ovis* (HF586911.1) sequences in the assay region.** The forward primer (218–242, yellow), reverse primer (272–294, blue), and probe (244–270, green) are all highlighted. A single nucleotide mismatch at probe position 225 (boxed in red) distinguishes *T. trichiura* from *T. vulpis* and *T. ovis*.
(PDF)

**S5 Fig. In silico alignment of *Plasmodium falciparum* (GenBank MN852864.1) and *Plasmodium relictum* (PV628049.1) sequences in the assay region.** The forward primer (686–708, yellow), reverse primer (765–785, blue), and probe (716–735, green) are all highlighted. There are no mismatches distinguishing *P. falciparum* from *P. relictum* in these regions.
(PDF)

## Acknowledgments

The authors gratefully acknowledge the property and business owners in Southern Indiana who provided us access to collect the samples used in this study.

## Author contributions

**Conceptualization:** Anna A. Heintzman, Drew Capone.

**Data curation:** Anna A. Heintzman.

**Formal analysis:** Anna A. Heintzman, Ishi Keenum.

**Funding acquisition:** Drew Capone.

**Investigation:** Anna A. Heintzman.

**Methodology:** Anna A. Heintzman, Drew Capone.

**Project administration:** Anna A. Heintzman.

**Supervision:** Drew Capone.

**Writing – original draft:** Anna A. Heintzman.

**Writing – review & editing:** Ishi Keenum, Drew Capone.

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
