## [Decision Letter · Decision Letter 0]

13 Jan 2026

Dear Dr. Capone,

Thank you for submitting your manuscript to PLOS ONE. After careful consideration, we feel that it has merit but does not fully meet PLOS ONE’s publication criteria as it currently stands. Therefore, we invite you to submit a revised version of the manuscript that addresses the points raised during the review process.

We look forward to receiving your revised manuscript.

Kind regards,

Marcello Otake Sato, Ph.D., D.V.M.

Academic Editor

PLOS One

Journal Requirements:

2. We note you have included a table to which you do not refer in the text of your manuscript. Please ensure that you refer to Table 1 in your text; if accepted, production will need this reference to link the reader to the Table.

“This work was supported by the National Science Foundation (Award # 2412444 to DC). This research was also funded by the Indiana University Advanced Summer Research Scholarship (2024 award to AH).”

Additional Editor Comments:

The manuscript by Heintzman, Keenum, and Capone addresses an interesting and underexplored area of zoonotic research, particularly within developed countries. The reviewers and the editor raised several minor points that should be addressed in a revised version.

First, concerns were noted regarding the title and the use of the term “pathogen.” As multiple species are examined, organisms identified as pathogens in one host species may act as commensals or symbionts in another. Therefore, the terminology should be clarified or adjusted as appropriate throughout the manuscript.

In addition, the explanation of the sample size calculation requires further clarification and methodological detail.

Finally, several issues related to the presentation of tables and figures were identified and should be carefully addressed by the authors in the revised manuscript.

Reviewers' comments:

Reviewer's Responses to Questions

**Comments to the Author**

1. Is the manuscript technically sound, and do the data support the conclusions?

Reviewer #1: Yes

Reviewer #2: Partly

2. Has the statistical analysis been performed appropriately and rigorously?

Reviewer #1: Yes

Reviewer #2: Yes

3. Have the authors made all data underlying the findings in their manuscript fully available?

Reviewer #1: Yes

Reviewer #2: Yes

4. Is the manuscript presented in an intelligible fashion and written in standard English?

Reviewer #1: Yes

Reviewer #2: Yes

Reviewer #1: The manuscript is well-structured and clearly written. The Methodology section (Materials and Methods) is consistent with the presented Results. All perceived contradictions (e.g., the diagnosis of Ascaris lumbricoides in swine) were appropriately addressed and clarified within the Discussion section.

General Recommendations for Figures and Tables

My primary suggestion is to implement minor revisions to the titles and captions of all main figures and tables to ensure they are self-explanatory and can be fully understood without referencing the main text.

As a general guideline, all figure and table titles/captions should be concise, descriptive, and contain the following essential components:

What is it? (The variables/data being presented).

Context/Subject of the Data. (The sample source/population).

Origin of the Data. (Location/Setting of the study).

Period in which the samples or data were obtained.

Additionally, the Notes section of each table/figure should briefly define all symbols, abbreviations, or statistical tests used.

For instance, the current title "Table 1. Prevalence of molecular targets" is excessively brief and requires expansion to meet these criteria.

Specific Observations on Supporting Information

Supporting Information / Supplementary Materials.docx

Table S2 & Table S3: Please revise the titles and notes according to the general recommendation above. Ensure the titles explicitly state the 'What,' 'Context,' 'Origin,' and 'Period.' The notes must clearly explain all symbols, abbreviations (e.g., TAC), and statistical analyses.

Table S4: The table number appears to be repeated in the manuscript draft, and the title remains extremely brief. Please correct the numbering and specify the exact object or assay system that was used for these tests.

Table S6: The same observations apply; please implement the general recommendations regarding the comprehensive title and notes.

Figure S2: I recommend omitting this figure. Its resolution is insufficient, and microscopic images of Ascaris sp. eggs are considered common knowledge within the field, being readily available in high quality in classic parasitology textbooks.

Table S7: Please complete the title information by including the location where the samples were obtained.

The term "Ascaris suum" should be replaced with "Ascaris type" or a similar designation. This is necessary because the manuscript acknowledges that "Eggs of A. suum and A. lumbricoides are morphologically indistinguishable" (Lines 298-299), thus preventing an unsupported species-level diagnosis based solely on morphology.

These constitute my only observations on the manuscript.

Reviewer #2: Please consider the following points:

1. The study objective should be clearly stated.

2. Regarding the research methodology, aspects such as sample size require explanation. Please specify the method used to determine the number of subjects.

**Do you want your identity to be public for this peer review?** For information about this choice, including consent withdrawal, please see our Privacy Policy

Reviewer #1: No

Reviewer #2: No

---

## [Author Response · Author response to Decision Letter 1]

20 Jan 2026

See uploaded response to reviewers document. We thank the reviewers for their thoughtful comments.

---

## [Editor Report · Decision Letter 1]

22 Jan 2026

Diverse enteric bacterial, viral, and parasitic pathogen genes are shed in animal feces in Indiana

PONE-D-25-54640R1

Dear Dr. Capone,

We’re pleased to inform you that your manuscript has been judged scientifically suitable for publication and will be formally accepted for publication once it meets all outstanding technical requirements.

Kind regards,

Marcello Otake Sato, Ph.D., D.V.M.

Academic Editor

PLOS One

Additional Editor Comments (optional):

I suggest to rephrase the sample calculation as: "No formal sample size calculation was performed, as the study was designed to provide a broad exploratory assessment of enteric pathogens and associated genes in animal fecal samples." in the subsequent stages of proofreading.

---

## [Editor Report · Acceptance letter]

PONE-D-25-54640R1

PLOS One

Dear Dr. Capone,

I'm pleased to inform you that your manuscript has been deemed suitable for publication in PLOS One. Congratulations! Your manuscript is now being handed over to our production team.

Kind regards,

on behalf of

Dr. Marcello Otake Sato

Academic Editor

PLOS One